# Stretchable Nanofiber-Based Felt as a String Electrode for Potential Use in Wearable Glucose Biosensors

**DOI:** 10.3390/s24041283

**Published:** 2024-02-17

**Authors:** Bianca Seufert, Sylvia Thomas, Arash Takshi

**Affiliations:** Department of Electrical Engineering, University of South Florida, 4202 E. Fowler Avenue, Tampa, FL 33620, USA; bseufert@usf.edu (B.S.); sylvia@usf.edu (S.T.)

**Keywords:** electrospun nanofibers, flexible string electrodes, wearable electronics, electrochemical glucose biosensor, cyclic voltammetry (CV), electrochemical impedance spectroscopy (EIS)

## Abstract

Nanofiber technology is leading the revolution of wearable technology and provides a unique capability to fabricate smart textiles. With the novel fabrication technique of electrospinning, nanofibers can be fabricated and then manufactured into a durable conductive string for the application of smart textiles. This paper presents an electrospun nanofiber mesh-based (NF-Felt) string electrode with a conducting polymer coating for an electrochemical enzymatic glucose sensor. The surface area of a nanofiber matrix is a key physical property for enhanced glucose oxidase (GOx) enzyme binding for the development of an electrochemical biosensor. A morphological characterization of the NF-Felt string electrode was performed using scanning electron microscopy (SEM) and compared with a commercially available cotton–polyester (Cot-Pol) string coated with the same conducting polymer. The results from stress–strain testing demonstrated high stretchability of the NF-Felt string. Also, the electrochemical characterization results showed that the NF-Felt string electrode was able to detect a glucose concentration in the range between 0.0 mM and 30.0 mM with a sensitivity of 37.4 μA/mM·g and a detection limit of 3.31 mM. Overall, with better electrochemical performance and incredible flexibility, the NF-Felt-based string electrode is potentially more suitable for designing wearable biosensors for the detection of glucose in sweat.

## 1. Introduction

With the recent advancements in semiconductor technology, there are now a myriad of low-power and compact electronic devices available for use in wearable electronics. These applications can be found across all industries such as entertainment, military, recreational activity, and medical monitoring. For example, a truly novel advantage of wearable biosensors is the ability to continuously monitor the vitals of active military personnel in extreme environments or the health of critical care patients. This work particularly focuses on string-shaped electrodes that have the potential to be suitable for designing wearable glucose sensors. Glucose sensors are used by millions of people every day to monitor the medical condition known as diabetes [1]. While most commercial glucose sensors utilize bio-analyte blood, a non-invasive glucose sensor would be a preferable alternative. Recent studies show the feasibility of detecting glucose in sweat, which opens doors to designing wearable biosensors for continuous monitoring in a safe way. For example, researchers are utilizing various nanomaterials such as gold nanosheets [2], porous enzymatic nanomembranes [3], and ZnO nanoparticles [4], all of which have seen the successful detection of glucose at low volumes in sweat-like analytes. Even more impressive is that they were all used in a scaled-down portable setting. This further highlights the advantages of utilizing an electrochemical approach for detection as compared with colorimetric or fluorometric approaches, which require slightly more advanced and time-consuming measurement tools to observe the emission or adsorption of energy.

Moreover, the rigid structure of commercially available glucose sensors is not suitable for wearable applications. A possible solution to overcome this challenge is through designing fibrous-based, string-shaped electrodes that can be sewn into a garment. Glucose sensors, in particular, need a porous structure to maximize the liquid interface with the electrode’s surface. Fibrous structures offer a large surface area, high porosity, and dimensionality advantages, which lead to enhanced binding of glucose enzymes for advanced and efficient electron transfer, allowing for higher sensitivity. To maximize the surface area, strings can be fabricated using nanofibers. Nanofibrous structures, in particular, show incredibly enhanced advantages in this area and can also be used in a variety of applications like flexible electrochemical transistors [5,6], filters [7], supercapacitors [8], and other forms of wearable sensors [9]. A reliable method to produce nanofibers is to use the fabrication technique known as electrospinning [10]. As shown in Figure 1, electrospinning a polymer on a flat collector produces a mesh structure [11]. This is due to the fact that a Taylor cone is induced when electrospinning and is critical to the definition of a nanofiber-based mesh [12]. While there are methods for developing aligned nanofibers during the electrospinning process [13], for the purpose of this paper, it was imperative to preserve the mesh structure and its critical surface geometry. Furthermore, to convert a nanofiber mesh into a string, the mesh must be rolled manually, forming a single-ply thread with a felt-like structure. In this work, the fabricated novel nanofiber-based felt (NF-Felt) string is compared with a commercially available thread. Due to the non-conductive nature of both threads, a piece of each was coated with a conducting polymer (poly(3,4-ethylenedioxythiophene) polystyrene sulfonate, known as PEDOT:PSS) and then glucose oxidase (GOx) was incubated as the enzyme for glucose detection. Given the unique nature of the phenomenon utilized during the electrospinning process, the threads considered for this work were highly scrutinized. Careful consideration regarding the durability, flexibility, and conductivity of the resulting fabricated electrodes is imperative to the development of this work and provides the researchers with insight into the possible electron transfer occurring along the surface of the electrode to liquid interface [14].

Overall, the synergistic approach to applying a wearable electrode to a biosensor continuously monitoring for glucose detection is no small feat, and current researchers are exploring this endeavor. Furthermore, with the growth occurring in this field, it is of no surprise that this unique technology has applications in smart textiles given the maximized surface area a nanofiber can provide in facilitating a direct electron transfer between the conducting electrodes. The work outlined here applies the discussed technology to an electrochemical cell for the advancement of biosensor research in flexible, non-invasive glucose-sensing applications.

## 2. Materials and Methods

### 2.1. Materials

High-conductivity-grade PEDOT:PSS (4% in H_2_O), D-Glucose (180.16 g/mol), potassium ferricyanide (329.26 g/mol), N,N-dimethylformamide (DMF), dodecylbenzenesulfonic acid (DBSA), ethylene glycol (EG), and polyvinylidene fluoride-co-hexafluoropropylene (PVDF-HFP) with the molecular weight of ~534,000 were purchased from Sigma Aldrich. The glucose oxidase (GOx) was from Aspergillus Niger.

### 2.2. Equipmental Procedure

Two different strings were used to fabricate the electrodes for this work. One utilized the commercially available (from Walmart, Bentonville, AR, USA) 25% cotton + 75% polyester blended string that is referenced to as the Cot-Pol-based string electrode. The other utilized a string based on fabricated PVDF-HFP nanofibers (NFs) that is referred to as the NF-Felt-based string electrode due to the formation of the nanofibers in a felt-like pattern. The polymer PVDF-HFP was selected due to its insolubility in water and enhanced mechanical stability under strain, making it a preferred choice when comparing it against commercially available textiles. Using an in-house designed electrospinning system, the PVDF-HFP NFs were fabricated by creating a 15 wt.% solution of PVDF-HFP in DMF and electrospinning under 22 kV at a distance of 20 cm onto a foil substrate using a 22 G needle at a rate of 0.2 mL/h for a total time of 5 min. The electrospun NFs were collected as a standalone mesh with an approximate mesh diameter of 5 cm. Then, using a rubber mat and a latex-based roller, the electrospun nanofiber mesh was rolled into a string. As shown in Step 2 of Figure 1a and the picture of the process in Appendix A, to convert the mesh into a string, the roller was manually rotated upward when it was pushed along the collector plate. The resulting string was then left in an isopropyl bath for 10 min to remove any excess solvent present on the NFs. Once rinsed and subsequently dried, the strings were then placed in the conductive PEDOT:PSS-based coating solution.

The conductive PEDOT:PSS-based coating solution was created by mixing PEDOT:PSS, DBSA, and EG with the 75:20:5 mass ratio following a recipe developed in our laboratory [15]. To create the working electrodes (WEs), a 3 cm long string of each type of material (i.e., NF-Felt and Cot-Pol) was individually submerged in the PEDOT:PSS coating solution. After 30 min, the strings were removed and left to dry for 8 h. This coating process was repeated twice for each string. The optical images of both strings under a digital microscope camera are shown in Figure 1d–g. Following the recipe from Aleeva et al. [16], once finally dried, each electrode was submerged in 100 µL GOx enzyme solution (50 mg/mL in DI water) for 30 min at 4 °C in order to immobilize the enzymes onto the surface of the string electrodes. Afterward, the electrodes were rinsed with the electrolyte solution to remove excess materials.

For the electrochemical experiments, a platinum wire was used as the counter electrode (CE) and a silver/silver chloride (Ag/AgCl) reference electrode (RE) was used in the electrochemical cell that was set up in a beaker. Following the recipe from Puttananjegowda et al. [17], the electrolyte solution was prepared using a 1:1 ratio of 5 mM potassium ferricyanide in sodium acetate buffer solution with a resulting pH of 7.2. Scanning electron microscope (SEM) images were obtained using the Hitachi SU-70 SEM (High-Tech, Schaumburg, IL, USA). Energy-dispersive X-ray spectroscopy (EDS) was performed using a Quanta 200 3D Dual Beam Electron Microscope (FEI Company, Hillsboro, OR, USA). To test the electrochemical responses to glucose, a 1.0 M glucose solution (in DI water) was prepared and added to the electrolyte solution to make the desired concentrations. The cyclic voltammetry (CV), electrochemical impedance spectroscopy (EIS), and chronoamperometry measurements were carried out using a VersaSTAT 4 Potentiostat (Princeton Applied Research (PAR), Oak Ridge, TN, USA). For the CV experiments, the scan rate was set at 50 mV/s and experiments were then conducted for 5 cycles. The presented CV loops in this work are from the third cycle of each experiment. EIS was conducted at a frequency range of 0.01 Hz–1 kHz by applying a sinusoidal signal with an amplitude of 50 mV in a 3-probe configuration. Chronoamperometry was performed by applying a constant voltage of 0.5 V to the working electrode vs. reference and by monitoring the current while a diluted glucose solution was added every 30 s to the electrolyte of the Cot-Pol electrode. The diluted glucose solution was added every 50 s for the NF-Felt electrode. The stress vs. strain test was conducted using the AILIGU XP-5N Digital Dynamometer (Ailigu, Shenzhen, China).

## 3. Results and Discussion

Our earlier studies on various commercially available threads have shown that the Cot-Pol string has a better conductivity than other tested threads when being coated with PEDOT:PSS [15]. Hence, we have used the Cot-Pol string coated with PEDOT:PSS as a reference to the novel string electrode that we have devised using electrospun nanofibers. SEM images of both strings before and after coating with PEDOT:PSS can be seen in Figure 2. Cot-Pol is a multi-ply string made from aligned twisted fibers with an average diameter of ~15 µm, forming a string with an overall diameter of ~270 µm. Both the optical and SEM images in Figure 1e and Figure 2b clearly show that the PEDOT:PSS coating is not very uniform on this string, perhaps due to the hydrophobicity of the Cot-Pol string. In contrast, the SEM images of the NF-Felt string show that the structure of this string is fundamentally different as the electrospun nanofibers form a mesh structure with an average fiber diameter of 225 nm (Figure 2c) and then become an NF-Felt-based string after rolling the mesh. This resulted in a single-ply string with an average diameter of ~520 µm. Figure 2e clearly shows that the NF-Felt has become saturated with the PEDOT:PSS solution, resulting in a conductive string when coated. Furthermore, in Figure 2g, only a slight amount of fluorine is present from the EDS analysis due to the PVDF-HFP nanofibers’ base of the observed NF-Felt string electrode. The aluminum present in the sample holder can be disregarded and the resulting C-, O-, and S-mapped results demonstrate a clear conductive coating across the entire length of the string. The mass of the strings (3 cm long) before and after coating with PEDOT:PSS was measured to estimate the amount of loaded PEDOT:PSS on each electrode. While the amount of loaded PEDOT:PSS was significantly higher on the Cot-Pol electrode (1.9 mg) than on the NF-Felt electrode (0.5 mg), the two-probe resistance measurement showed 100.6 kΩ/cm in the Cot-Pol electrode and 89.3 kΩ/cm in the NF-Felt string electrode. Considering the lower resistance in the NF-Felt for a much smaller amount of loaded PEDOT:PSS, it is evident that the structure of the NF-Felt string is affecting the conductivity of the string electrode.

In addition to the physical differences between the two strings, their mechanical strength was also quite different. Figure 3 shows the stress–strain responses from both electrodes before and after being coated with PEDOT:PSS. As previously reported [15], the tensile strength in the Cot-Pol thread improved after coating with the conducting polymer. Comparing the string flexibilities, the NF-Felt string was found to be far more stretchable than Cot-Pol, reaching 60% strain before breaking. Although the coating does show a slight brittleness and therefore impacts slightly on the stretchability of the NF-Felt string electrode, the conductive NF-Felt was still more stretchable than the other electrode. Additionally, the low Young’s modulus in the NF-Felt electrode (19.3 MPa versus 3.47 GPa in Cot-Pol) clearly shows the mechanical stretchability of the string.

Furthermore, both string electrodes were electrochemically characterized using a three-probe setup. Each PEDOT:PSS-coated string was electrochemically characterized in the K_3_Fe(CN)_6_ electrolyte in three steps, as follows: 1- before GOx immobilization (defined as “Baseline”); 2- after 30 min treatment at 4 °C in the GOx solution (defined as “Enzyme”); and 3- after adding glucose (30 mM final solution concentration) to the electrolyte (defined as “Glucose”). Figure 4 shows the CV and EIS responses of the Cot-Pol and NF-Felt string electrodes. Since the amount of loaded PEDOT:PSS is different in the Cot-Pol and the NF-Felt strings, to compare their CV and chronoamperometry results, the current was normalized to the loaded amount of PEDOT:PSS and expressed with the unit of mA/g. 

As seen in Figure 4, there are defined peaks for both Cot-Pol and NF-Felt string electrodes elaborating upon the redox reactions occurring at the electrodes’ surfaces ([Fe(CN)_6_]^4−^ ↔ [Fe(CN)_6_]^3−^ + e^−^). While from the CV results (Figure 4a), it is evident that the double-layer capacitive behavior of the Cot-Pol electrode is dominant in all three loops, the results from the NF-Felt (Figure 4b) show clear redox peaks. At baseline, the NF-Felt shows an oxidation peak around +0.3 V and a reduction peak around −0.4 V. However, after the enzyme immobilization, a new pair of redox peaks appeared, confirming that the immobilized GOx can effectively interact with the string electrode. Both peaks shifted significantly after introducing glucose to the electrolyte. As shown in the inset of Figure 4b, the chain of redox reactions with glucose includes the conversion of glucose to gluconic acid while reducing GOx. The reduced GOx oxidizes by giving an electron to [Fe(CN)_6_]^3−^ to convert it into [Fe(CN)_6_]^4−^. In the third reaction, ferrocyanide delivers an electron to the electrode and converts back to ferricyanide ([Fe(CN)_6_]^4−^ → [Fe(CN)_6_]^3−^ + e^−^) [18]. As is more clearly shown in Appendix A, the CV results from both electrodes verify a strong redox reaction at the surface of the NF-Felt electrode. However, due to the capacitive response of the Cot-Pol electrode, the redox peaks are weaker than those in the NF-Felt string electrode. Also, increasing the scan rate in the CV experiments for the NF-Felt electrode (Appendix A) shows an expansion of the voltage between the oxidation and reduction peaks, confirming the electron transfer process when glucose reacts with GOx in the NF-Felt electrode. The electron transfer nature is also evident in the EIS results from the NF-Felt electrode. While the low-frequency tails in the EIS of the Cot-Pol electrode in Figure 4c suggest a domination of diffusion, the semicircle shape of EIS for the NF-Felt string in Figure 4d emphasizes the domination of the electron transfer mechanism in the stretchable string. Knowing that the diameter of the semicircle (R_ct_) is proportional to the inverse of the electron transfer rate, the results from Figure 4d show that after immobilizing Gox, the electron transfer significantly improved. This can be interpreted as an effective immobilization of GOx on the surface of the NF-Felt electrode. However, we observed a reduction in the electron transfer rate when glucose was introduced to the electrolyte. Considering the chain reactions through three cycles of redox reactions (Figure 4b’s inset), the overall electron transfer rate to the electrode is expected to be limited by the slowest reaction in the chain. 

To assess the response of each electrode to the variation of glucose, chronoamperometry experiments (amperometric tests) were conducted by applying 0.5 V to the working electrode and by monitoring the current while a glucose solution was added in ten steps, increasing the glucose concentration by 3 mM at each step, as seen in Figure 5. To clarify, exactly 20 µL of a 1.0 M D-Glucose solution was added to the electrochemical cell every 30 s to increase the concentration of glucose present in the electrolyte to then be measured by the Cot-Pol string electrode. The same approach was performed every 50 s for the NF-Felt string electrode. Given that there was 6.5 mL of electrolyte present in the cell, this increased the concentration by 3 mM at each addition of 20 µL of glucose (i.e., the “step” seen in Figure 5a,b). The NF-Felt string electrode response was consistent with a 0.17 mA/g increase after each additional drop of the glucose solution (Figure 5b). Furthermore, the response between each step remained consistently around 4 s. The same cannot be said for the Cot-Pol thread (Figure 5a). While initially responding within 3 s for a 60 µA/g increase, the Cot-Pol string electrode’s ability to consistently respond over time was unsuccessful, resulting in a 10 s response time with a 0.15 mA/g increase as the string was tested at a 24 mM concentration of glucose. It is important to note that over the chronoamperometry testing period, the electrolyte solution for the Cot-Pol string electrode did slowly start to turn from a yellowish hue to a blueish green, signaling that the PEDOT:PSS coating was starting to come off, possibly due to the poor adhesion of PEDOT:PSS to the string. This could potentially explain why the consistency of the response to glucose could not be maintained at the surface of the Cot-Pol string electrode. In contrast, the NF-Felt electrode demonstrated a remarkable stability and consistent response to the addition of 3 mM glucose. 

The results from the chronoamperometry tests were further analyzed in Figure 5c,d to find the sensitivity and the limit of the detection (LOD) using Microsoft Excel software, https://www.microsoft.com/en-us/microsoft-365/excel. Based on the measured current density versus the glucose concentration, the linear regression calibration curve was calculated for each electrode by applying a linear trendline, as seen in Figure 5c,d. The value of the correlation coefficient (R^2^ = 0.9936) for the NF-Felt electrode clearly shows the linear response of the sensor. From the slope of the calibration curve, the sensitivity was calculated for each electrode and, using the same method reported previously [18], the limit of detection (LOD) was calculated (LOD = 3.3 × *Sy*/*S* where *Sy* is the standard deviation from the fitted linear curve and *S* is its slope). The results from the NF-Felt sample show higher sensitivity (37.4 μA/mM·g) and better LOD (3.3 mM) than those in Cot-Pol (sensitivity = 32.1 μA/mM·g and LOD = 12.7 mM). To compare our results with others, a few of the recent publications with flexible electrodes for glucose sensing are listed in Table 1. In other works, the sensitivity was reported based on the normalized measured current to the surface area of the electrodes; we estimated the apparent surface area of the NF-Felt string electrode (assuming a cylinder with a diameter of 0.05 cm and length of 3 cm) and used it to calculate the sensitivity. Comparing the results, the sensitivity and LOD of the NF-Felt electrode are in the same range as other recently reported devices. Considering the unique stretchability and flexibility of the NF-Felt string, the new electrode design is potentially suitable to be used for developing a wearable glucose sensor that may be sown near the armpit area (which requires a high level of flexibility and stretchability) of a patient’s garment for monitoring glucose from the patient’s sweat. While these results are promising, the NF-Felt string electrode requires testing to ensure durability over time for either repeated or one-time use. More importantly, to design a wearable sensor device, further work is needed to design string-shaped counter and reference electrodes to integrate all three electrodes into a fabric structure. The wearable sensor then must be characterized for selectivity to the chemicals in sweat and checked for performance for positive and negative controlled samples before being used in real applications. Additionally, the stability and repeatability of the sensor response should be verified for accurate measurements. 

## 4. Conclusions

In conclusion, a Cot-Pol string electrode was used as a comparison flexible electrode in an effort to define the capabilities of a nanofiber mesh-based (NF-Felt) string electrode. The results show that the fiber structure within both electrodes clearly plays a role in the direct electrode interfacing with the conductive polymer coating. While the Cot-Pol response was dominated by the diffusion of ions, establishing a double layer charge at the electrode-electrolyte interface, the electron transfer process, which is directly correlated to glucose detection, was apparently promoted by the nanostructure of the NF-Felt string. Also, the NF-Felt string was far more flexible and stretchable than the Cot-Pol string, making it far more suitable for designing a wearable sensor. Furthermore, it is important to address that overall, the NF-Felt string electrode provided a stable response to the addition of glucose within the electrolyte over time. Applications for a flexible, stretchable biosensor are wide and varied throughout the medical and military fields. The beneficial use of NF-Felt string electrodes in smart materials for athletic apparel for constant health monitoring is also promising. The insights found within this work contribute to the development of flexible materials for biosensor research and hopefully, one day, can bring in a new generation of flexible integrated electronics.

## Figures and Tables

**Figure 1 sensors-24-01283-f001:**
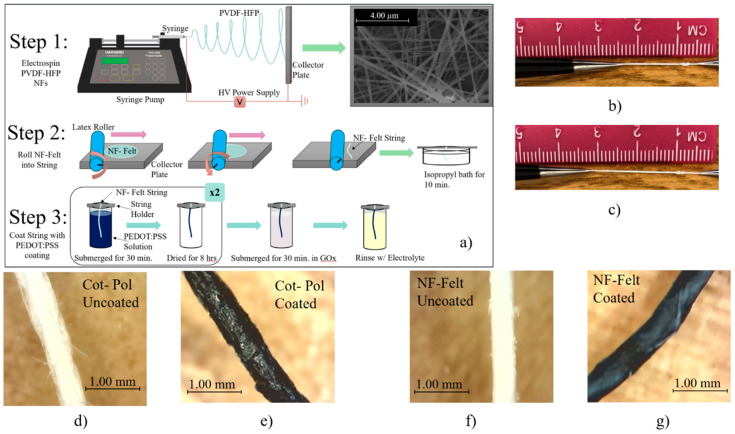
(**a**) Diagram of the NF-Felt string fabrication procedure. NF-Felt string (**b**) at rest and (**c**) being stretched. Cot-Pol string (**d**) before and (**e**) after coating with PEDOT:PSS. NF-Felt string (**f**) before and (**g**) after coating with PEDOT:PSS.

**Figure 2 sensors-24-01283-f002:**
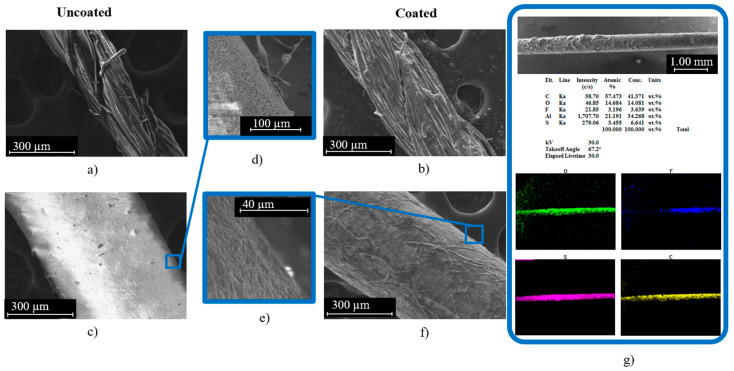
(**a**) Cot-Pol string for fiber observation (uncoated) at 300 µm; (**b**) Cot-Pol string for string observation (coated); (**c**) NF-Felt string for fiber observation (uncoated) at 300 µm; (**d**) surface close up of uncoated NF-Felt; (**f**) NF-Felt string for string observation (coated); (**e**) surface close up of coated NF-Felt; (**g**) EDS analysis of coated NF-Felt electrode.

**Figure 3 sensors-24-01283-f003:**
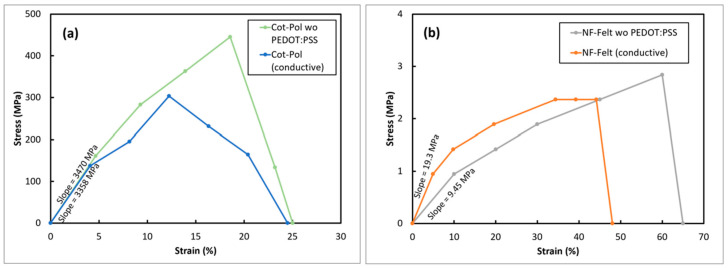
Stress vs. strain responses from (**a**) Cot-Pol and (**b**) NF-Felt string electrodes.

**Figure 4 sensors-24-01283-f004:**
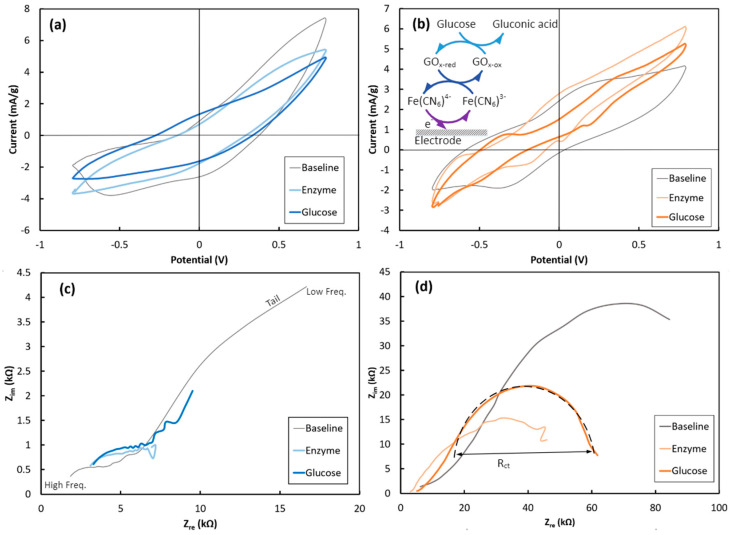
Cyclic voltammetry responses of the (**a**) Cot-Pol and (**b**) NF-Felt string electrodes. Electrochemical impedance spectroscopy responses of the (**c**) Cot-Pol and (**d**) NF-Felt string electrodes (inset) and the chain reactions associated with the glucose detection.

**Figure 5 sensors-24-01283-f005:**
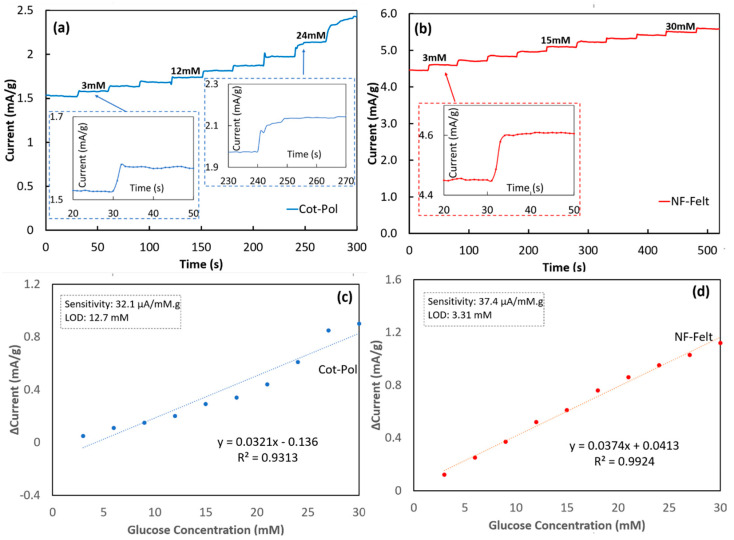
Chronoamperometry test responses of (**a**) Cot-Pol string and (**b**) NF-Felt electrodes. Δ Current (signal current at zero glucose concentration) vs. glucose concentrations for (**c**) Cot-Pol and (**d**) NF-Felt electrodes.

**Table 1 sensors-24-01283-t001:** Comparison of NF-Felt-based string electrode with other PEDOT-based approaches for glucose sensing.

Reference	Sensing Electrode	Sensitivity	Limit of Detection	Concentration Range
[18]	Au/SiCNPs-PEDOT:PSS-PVDF-ENFM/GOx	30.75 μA/mM cm^2^	3.825 mM	0.5–20 mM
[16]	Pt/PEDOT-NFs/GOx	9.2 μA/mM cm^2^	0.26 mM	0.1–25 mM
[17]	Au/PEDOT:PSS-PVDF-ENFM/GOx	5.11 μA/mM cm^2^	2.3 μM	0–25 mM
[19]	PEDOT-NFs/Gox-3	74.22 μA/mM cm^2^	2.9 μM	0–4.5 mM
**This Work (NF-Felt)**	**NFs/PEDOT:PSS**	**37.4 μA/mM g** **(0.386 µA/mM cm^2^)**	**3.31 mM**	**0–30 mM**

## Data Availability

Data will be made available upon request.

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
