# Peer review of "Stretchable Nanofiber-Based Felt as a String Electrode for Potential Use in Wearable Glucose Biosensors"

_sensors, 2024, doi:10.3390/s24041283_

Round 1
Reviewer 1 Report
Comments and Suggestions for Authors
The authors described electrochemical method for detection of glucose based on stretchable nanofiber as a wearable biosensor. Although the manuscript is interesting but some comments should be considered for improvement.
Although the conducted study is based on electrochemical biosensor but the other applied methods based on colorimetric and fluorometric studies should be discussed.
The connection between this approach and the electrochemical signal isn't adequately established.
Furthermore, what is concerning is the real samples used to test the method efficacy. There is no mention of how this test was performed, on which samples, from where these were taken, if the authors have checked the performance of the explained methodology with positive and negative controls.
Comments on the Quality of English Language
Minor editing of English language is required.
Author Response
Thank you for your comments. All the comments are addressed point by point in the attached Report Note and applied in the revised manuscript.

Reviewer 2 Report
Comments and Suggestions for Authors
1- In EIS, why did the addition of enzyme and glucose lead to the increase and decrease of electron transfer, respectively? Explain in the text. The Fig. 4 should be further discussed.
2- Why the fabricated electrode has not been tested in real solution (e.g. sweat)?
3- How did you estimate the surface area of the bare and modified electrodes?
4- More characterizations of the synthesized samples are advised.
5- The use of abbreviations in the manuscript should be revised.
6- I would advise to separate the references so that the reader can know which reference corresponds to which part of the sentence, you have mostly referenced your text with stacked references.
7- The use of units in the text should be checked.
8- The insets in Fig. 5 can be modified.
Comments on the Quality of English Language
There are typos in the manuscript. Hence the editing of the English language is advised.
Author Response
Thank you for your comments. All the comments are addressed point by point in the attached file and applied in the revised manuscript.

Reviewer 3 Report
Comments and Suggestions for Authors
see the attached

Author Response

(The authors gave the same response as above.)

Reviewer 4 Report
Comments and Suggestions for Authors
This paper presents an electrospun nanofiber mesh-based (NF-Felt) string electrode with a conducting polymer coating for an electrochemical enzymatic glucose sensor. The author claimed that the NF-Felt string electrode is suitable for the detection of glucose in sweat. However, the manuscript doesn’t provide sufficient characterizations to support their conclusions. The specific questions and comments are listed as below:
1. The title of this paper is ‘Stretchable Nanofiber Based Felt as a String Electrode for Wearable Glucose Biosensors’, which means the as-prepared string electrode can be used as a wearable device. However, the relevant supporting data, such as detection selectivity, anti-interference, repeatability, and stability was not provided.
2. In this study, the reason why polyvinylidene fluoride-co-hexafluoropropylene (PVDF-HFP) was used for fabrication of nanofiber-based string electrode was not explained. In addition, the as-prepared string electrode should be further characterized, such as EDS, FTIR, etc.
3. In Figure 4, the reaction which occurred at the surface of electrode should be indicated clearly.
4. The author claimed that the NF-Felt string electrode can detect a glucose concentration in a range of 0.0 to 5.0 mM in the part of Abstract, but the concentration range was expressed as 0 to 30 mM in Table 1. The statement was not consistent.
5. In page 7, line 232, ‘we have estimated the apparent surface area of the NF-Felt string electrode’, the calculation method should be provided.
6. In Figure 4a, b and Figure 5, the unit of current was expressed as ‘μA/g’.
7. To enable readers more clearly understanding the study, I suggest that the Results and Discussion should be combined.
Comments on the Quality of English LanguageThe English language is clearly presented.
Author Response

(The authors gave the same response as above.)

Round 2
Reviewer 3 Report
Comments and Suggestions for Authors
The glucose concentration measurement results is still not available. It is suggested that the authors should finish the complete measurement and calculations if they want to publish the paper on this journal.
Comments on the Quality of English Languageproper modification is required
Author Response
A file is attached.

Reviewer 4 Report
Comments and Suggestions for Authors
I think the revised version has solved most of the issues in the last version, I recommend to accept.
Author Response
Thank you for your recommendation for accepting the manuscript for publication.